# Single-Position Oblique Lumbar Interbody Fusion and Percutaneous Pedicle Screw Fixation under O-Arm Navigation: A Retrospective Comparative Study

**DOI:** 10.3390/jcm12010312

**Published:** 2022-12-30

**Authors:** Hyung Cheol Kim, Yeong Ha Jeong, Sung Han Oh, Jong Min Lee, Chang Kyu Lee, Seong Yi, Yoon Ha, Keung Nyun Kim, Dong Ah Shin

**Affiliations:** 1Department of Neurosurgery, Bundang Jesaeng General Hospital, 20, Seohyeon-ro 180 beon-gil, Bundang-gu, Seongnam-si 13590, Republic of Korea; 2Department of Neurosurgery, International St. Mary’s Hospital, College of Medicine, Catholic Kwandong University, 25, Simgok-ro 100 gil, Seo-gu, Incheon 54671, Republic of Korea; 3Department of Neurosurgery, Spine and Spinal Cord Institute, Severance Hospital, Yonsei University College of Medicine, 50-1, Yonsei-ro, Seodaemun-gu, Seoul 03722, Republic of Korea

**Keywords:** oblique lateral lumbar interbody fusion, minimally invasive surgery, O-arm navigation, C-arm, spinal fusion

## Abstract

The insertion of pedicle screws in the lateral position without a position change has been reported. We completed a retrospective comparison of the radiologic and clinical outcomes of 36 patients who underwent either single-position oblique lateral lumbar interbody fusion (SP-OLIF) using the O-arm (36 cases) or conventional OLIF (C-OLIF) using the C-arm (20 cases) for L2–5 single-level lumbar degenerative diseases. Radiological parameters were analyzed, including screw accuracy (Gertzbein-Robbins classification system; GRS), segmental instability, and fusion status. Screw misplacement was defined as a discrepancy of ≥2 mm. Clinical outcomes, including visual analog scale, Oswestry Disability Index (ODI), 36-Item Short Form Health Survey (SF-36), and postoperative complications, were assessed. The spinal fusion rate was not different between the SP-OLIF and C-OLIF groups one year after surgery (*p =* 0.536). The ODI score was lower (*p =* 0.015) in the SP-OLIF than the C-OLIF group. Physical (*p =* 0.000) and mental component summaries (*p =* 0.000) of the SF-36 were significantly higher in the SP-OLIF group. Overall complication rates, including revision, surgical site infection, ipsilateral weakness, and radicular pain/numbness, were not significantly different. SP-OLIF using the O-arm procedure is feasible, with acceptable accuracy, fusion rate, and complication rate. This may be an alternative to conventional two-stage operations.

## 1. Introduction

The oblique lumbar interbody fusion (OLIF) has gained popularity as a minimally invasive spinal fusion technique for the treatment of degenerative lumbar diseases including lumbar stenosis, spondylolisthesis, degenerative disc diseases, spinal instability, and spinal deformity. Compared to traditional posterior approaches, such as posterior lumbar interbody fusion (PLIF) and transforaminal lumbar interbody fusion (TLIF), OLIF provides favorable fusion by enabling a large fusion bed, and facilitates early recovery with less muscle damage, blood loss, and wound infection [1,2,3,4,5]. Unlike PLIF and TLIF, OLIF traditionally requires repositioning the patient from supine to prone for supplemental pedicle screw fixation.

Recently, the O-arm and intraoperative navigation techniques have become increasingly important in spinal surgery [6,7]. These techniques increase the accuracy of pedicle screw placement and cage insertion, and reduce malposition compared with freehand and conventional fluoroscopy techniques [8,9,10]. Fluoroscopy in the lateral position during pedicle screw insertion is awkward and inconvenient for surgeons. Because the O-arm system is removed from the operating field during virtual navigation, pedicle screws can be inserted from the lateral position without machine intervention. In addition, lower radiation exposure to surgeons and surgical teams is an advantage of the O-arm compared to the C-arm [11,12].

Previous studies on pedicle screw insertion in the lateral position without positional change have recently been reported [13,14,15,16,17]. Therefore, we considered that performing OLIF in a single position by taking advantage of the O-arm would be very efficient in various clinical aspects and can be helpful for surgeons who are just starting out with single-segment OLIF. We aimed to evaluate the accuracy and efficiency of the single-position OLIF (SP-OLIF) using the O-arm procedure and to obtain clinical evidence supporting SP-OLIF by comparing two surgical methods: SP-OLIF using the O-arm and conventional OLIF (C-OLIF) using the C-arm.

## 2. Materials and Methods

### 2.1. Patient Population

Between June 2017 and September 2020, 76 patients who underwent either SP-OLIF using the O-arm (49 cases) or C-OLIF using the C-arm (27 cases) for L2–5 single-level fusion for lumbar degenerative diseases were enrolled in this study. All patients provided written informed consent and the relevant Institutional Review Board of the Yonsei University Health System, Severance Hospital approved this study (4-2021-1528). Among the 76 patients who underwent L2-5 single level lumbar fusion surgeries, 56 patients (SP-OLIF: 36 cases vs. C-OLIF: 20 cases) who met the inclusion criteria or did not meet the exclusion criteria were analyzed retrospectively (Figure 1). There were 16 males and 20 females in the SP-OLIF group and six males and 14 females in the C-OLIF group. The inclusion criteria were as follows: age > 18 years, patients who underwent surgery at the level of L3–4 or L4–5, spinal stenosis with typical symptoms, degenerative or spondylolytic spondylolisthesis of Meyerding grade 1 or 2, failure of conservative treatment for more than six months, and available follow-up data for at least 12 months. The exclusion criteria were as follows: spinal infection, systemic malignancy, spinal trauma, previous spinal surgery, history of abdominal surgery, diagnosed osteoporosis, incomplete follow-up, or missing medical records. The average age at surgery was 61 years in both groups. The mean body mass index in each group was 25.6 and 25.0, respectively. The selection of surgery was not only based on the image findings, surgical period and requirement of direct neural decompression, but also the surgeon also decided on the surgical procedure according to the patient’s request and the surgeon’s discretion after explaining the pros and cons of each surgical procedure.

### 2.2. Data Collection

Variables, including demographic characteristics, disease-related data, parameters related to the operation, and outcomes, were investigated. Demographic parameters included sex, age, body mass index (BMI), bone mineral density (BMD), and the American Society of Anesthesiologists class. A T-score < −2.5 on dual-energy X-ray absorptiometry was defined as osteoporosis. Disease parameters included diagnosis, index levels, and Meyerding grade [18]. Clinical outcomes were routinely assessed using the visual analog scale (VAS), Oswestry Disability Index (ODI), and the 36-Item Short Form Health Survey (SF-36). All patients received follow-up X-rays at one, three, six, and 12 months, and computed tomography (CT) at 12 months. Radiographs were used to evaluate segmental instability. CT scans were used to determine the fusion status one year postoperatively. Information on the operation time, estimated blood loss (EBL), American Society of Anesthesiologists (ASA) score, and hospital stay was obtained from chart reviews.

### 2.3. Surgical Technique

The surgeries were performed by two spine surgeons with more than 20 years of surgical experience at our institution using the same protocol.

#### 2.3.1. SP-OLIF

Patients were placed in the 70° right lateral decubitus position on a radiolucent table to facilitate percutaneous pedicle screw fixation (PPSF) (Figure 2).

After preoperative administration of skin antiseptics, the surgical field was covered with an iodine-impregnated incision drape. A navigation reference arc was then placed over the iliac crest approximately 2 inches superolateral to the posterior superior iliac spine. Reference arrays were registered for real-time navigation (Stealth Station, Medtronic, Memphis, TN) based on the first CT scan (O-Arm, Medtronic, Memphis, TN). An oblique incision was made 3 cm anterior to the mid-portion of the index disc space. Blunt dissection was gently performed with sequential exposure of the external oblique, internal oblique, and transversus abdominis fascia. The retroperitoneal fat and space were then identified by visualizing the psoas muscle. The adventitial layers were mobilized at the anterior aspect of the psoas muscle with gentle dissection to allow a wider surgical corridor and to avoid injuring the psoas muscle and lumbar plexus. Navigation was used to identify the correct disc space for entry anterior to the psoas. Sequential dilators were inserted, and a tubular retractor and a light source were placed. A shaver, curette, trial implant, and cage (Clydesdale, Medtronic, Memphis, TN, USA) were used for navigation. The intervertebral cage was filled with the graft material (Grafton, Medtronic, Memphis, TN, USA). The cage was placed using an orthogonal maneuver to achieve a 90° angle during the cage placement. To confirm cage position and obtain a preoperative image for PPSF, a second CT scan was performed. The retractor was removed and the abdominal wound was closed in layers.

Before PPSF, the patients were positioned in the 70° right decubitus position to facilitate downside screwing (right side). Percutaneous screws (Legacy, Medtronic, Memphis, TN) were placed using the CD Horizon Sextant system and Stealth Navigation, without changing the patient’s position (Figure 3). Finally, a third CT scan was performed to confirm ideal pedicle screw and rod positioning.

#### 2.3.2. C-OLIF

In the C-OLIF procedure, the patient was placed in the right lateral decubitus position and a 2-inch skin incision was made in the same manner as SP-OLIF. Fluoroscopy using the C-arm (OEC 9900 Elite, General Electric Company, Boston, MA, USA) was performed to confirm the proper level and successful cage insertion. After cage placement, the patients were placed in the prone position for PPSF using the C-arm.

### 2.4. Radiologic Outcomes

A postoperative CT scan was used to analyze the accuracy of screw placement using the Gertzbein-Robbins classification system (GRS) immediately after surgery [19]. Plain radiographs, including anteroposterior, lateral, flexion, and extension views, were obtained before surgery and at six months and one year postoperatively to compare the segmental stability and fusion status. We defined a mobility of <5° at the index level as solid fusion. Conventional CT scans were obtained postoperatively at 12 months to assess bony fusion by confirming bridging trabecular bone (BTB) in sagittal and coronal cuts. To minimize inter- and intra-observer errors, two independent spine surgeons evaluated the plain radiographs and CT scans.

### 2.5. Clinical Outcomes

VAS and ODI were assessed preoperatively and at 12 months postoperatively. Pain intensity was reported from 0 to 10 using a subjective VAS (0 = no pain; 10 = worst pain imaginable). The ODI scores ranged from 0 to 100 (0 = no disability; 100 = maximum disability). All patients also completed the 36-Item Short Form Survey (SF-36), which consisted of a physical component summary (PCS) and a mental component summary (MCS). All clinical outcome scales were surveyed preoperatively and at one, three, six, and 12 months after surgery by a pain-specialist nurse who was blinded to the type of surgery. Postoperative complications, including wound revision, wound infection, postoperative pain, acquired weakness, and sensory changes, were investigated.

### 2.6. Statistical Analysis

The results are expressed as mean ± standard deviation. Categorical data are presented as numbers (%). A repeated measures analysis of variance was used to evaluate differences in the VAS and ODI scores between the SP-OLIF cage and C-OLIF groups before surgery and at one year after surgery. The fusion rate was calculated for each group, and between-group differences in the rates of fusion and postoperative complications were compared using the chi-square test or Fisher’s exact test, as appropriate for data distribution. All statistical analyses were performed using SPSS version 23 for Windows (IBM, Armonk, NY); *p* < 0.05 was considered statistically significant.

## 3. Results

### 3.1. Patient Demographics

The demographic characteristics of patients are presented in Table 1. There were no significant differences between the two groups regarding age, sex, BMI, BMD, operative time, ASA class, preoperative diagnosis, and instrumented level distribution. However, the hospital stays were significantly higher in the SP-OLIF group than that of C-OLIF group (7.97 ± 2.43 vs. 5.40 ± 1.00, *p =* 0.000). In the SP-OLIF group, the EBL was significantly less than in the C-OLIF group (131.94 ± 95.40 vs. 270.00 ± 238.64, *p =* 0.003).

### 3.2. Radiologic Outcomes

#### 3.2.1. Pedicle Screw Accuracy

Table 2 shows the pedicle screw placement accuracy measured using GRS. In the SP-OLIF group, 144 pedicle screws were inserted, while 80 pedicle screws were inserted in the C-OLIF group. There were no significant differences between the groups in distribution of each GRS grade. According to the GRS of SP-OLIF group, 139 screws (96.5%) were classified as grade A and five screws (3.5%) as grade B. There were no cases of grades C, D, or E. In the C-OLIF group, 77 screws (96.3%) were classified as grade A, three screws (3.8%) as grade B, one screw (1.3%) as grade C, and there were no cases of grade D or E (Table 3).

#### 3.2.2. Radiologic Parameters and Fusion Rates

Bone fusion and one-year postoperative Cobb angles measured using lateral flexion/extension plain films of the instrumented levels are summarized in Table 4. There were no differences in the fusion rates between the two groups at one year after surgery (94.4% vs. 90.0%, *p =* 0.536). Furthermore, the number of BTB formations was not significantly different between the groups (33/36 segments [91.7%] vs. 17/20 segments [85.0%], *p =* 0.930). There was no significant difference in radiologic parameters (the mean Cobb angle of the instrumented level at flexion, extension, and the amount of motion [flexion minus extension]) between the groups at one year after surgery.

### 3.3. Clinical Outcomes

Preoperative clinical outcomes and those one year after surgery are summarized in Table 5. The mean ODI at one year after surgery was lower (showing an improved outcome) for the SP-OLIF group than for the C-OLIF group (18.81 ± 10.99 vs. 28.20 ± 17.06, *p =* 0.015). Preoperative pain decreased significantly at one year after surgery (7.31 ± 1.31 vs. 2.56 ± 2.04, *p =* 0.002). In the SF-36 survey, the PCS (62.22 ± 14.09 vs. 47.63 ± 15.03, *p =* 0.000) and MCS (74.17 ± 13.82 vs. 52.89 ± 15.68, *p =* 0.000) of SF-36 were significantly higher in the SP-OLIF group than in the C-OLIF group.

### 3.4. Surgical Complications

Postoperative complications, including revision surgery, are summarized in Table 6. There were no cases of wound revision or infection in the SP-OLIF group. However, in the C-OLIF group, there was one revision case due to screw malposition.

There was one case (2.8%) of motor weakness in the SP-OLIF group, but there were no cases of postoperative weakness in the C-OLIF group. Foot drop occurred immediately after surgery but completely recovered two weeks later. The number of patients complaining of temporary radicular pain and numbness was three (8.3%) in the SP-OLIF group, and two (10%) in the C-OLIF group, but all these patients recovered after three months. All of the above-mentioned complications resolved after conservative treatment.

## 4. Discussion

Since Amiot et al. first reported the use of a computer-assisted navigation system for pedicle screw fixation in 1995 [20], navigation technology has been used in various spinal surgical procedures worldwide [21,22,23]. O-arm navigation produces high-quality images comparable to those of conventional C-arm scans and provides the surgeon with clear intraoperative guidance. Three-dimensional (3D) real-time navigation, performed using an intraoperative O-arm system, can reveal 3D anatomic structures. The more intuitive 3D-position guidance provides a significant advantage in complex spine surgery [24,25].

Since Mayer first reported OLIF in 1997, it has become one of the most popular minimally invasive surgical procedures worldwide [26]. OLIF spares the psoas and provides direct visualization of key structures, while minimizing the risk of injury to the lumbar plexus, ureter, and great vessels [27]. However, conventional OLIF requires the repositioning of the patient from the lateral position to the prone position during surgery. Repositioning is time-consuming and has the potential to increase perioperative risk [28]. To overcome this issue, several surgeons have attempted pedicle screw fixation in the lateral position, that is, SP-OLIF with PPSF [29]. Favorable outcomes have been reported with clinical feasibility. Blizzard reported that the pedicle screw breach rate was 5.1% and the fusion rate at six months postoperatively was 87.5% in SP-OLIF with PPSF under C-arm fluoroscopy [13]. Drazin et al. reported that the operation time was shorter in single-position lateral interbody fusion and PPSF than in repositioned patients (130 min vs. 190 min, *p =* 0.009) [30]. In our study, a shorter operation time and less EBL were observed in the SP-OLIF group than in the C-OLIF group, although the EBL was significantly different.

Pedicle screw placement accuracy was also high in this study, similar to the results of previous studies. Xi et al. reported that a total of 350 levels were operated upon using a navigation system, and 94.86% of cages were placed within the acceptable range [12]. Tian and Xu reported that CT-based navigation systems had higher accuracy rates for pedicle screw placement than fluoroscopic guided screws (90.76% vs. 85.48%) [31]. In a meta-analysis by Feng et al., O-arm navigation had significant advantages in terms of accuracy over conventional C-arm fluoroscopy [32]. In addition, several studies have evaluated the accuracy of PPSF under navigation guidance in the lateral position [12,33,34]. Ouchida et al. reported that the rate of screw misplacement in single-position OLIF using O-arm navigation is only 1.8% [14,33].

In contrast, Hiyama et al. raised the criticism that inserting screws while viewing fluoroscopy in a lateral position is unfamiliar to surgeons, and that a working space between the patient and fluoroscope cannot be secured [29]. In addition, Mills et al. reported that pedicle screws placed in the lateral position had a higher rate of complications than those placed in the prone position [16]. The oblique angle can also be disorienting for surgeons, and navigation may be a method to mitigate this disorientation and solve these problems [35]. To offset this disorientation, we reduced the angle by 20° in the right lateral position and set it to 70°. Spatial perception can be secured using real-time position tracking, even in the awkward lateral position. Furthermore, the O-arm was retrieved after scanning; therefore, the working space was much wider. In our study, no screw placements were outside the clinically acceptable range. We believe that the use of the O-arm provides not only a precise view of the anatomy in an unfamiliar position but also high screw insertion accuracy.

In our study, which used SP-OLIF, there was one case (2.8%) of motor weakness, and three patients (8.3%) showed temporary radicular pain and numbness, but all recovered after three months. Lateral cage misplacement has been reported to range from 0.26 to 3.8% [36]. However, our patient did not experience radicular pain due to cage misplacement. Radicular pain is thought to be caused by genitofemoral nerve irritation or nerve stretch caused by the elevation of the intervertebral space. Unlike lateral lumbar interbody fusion (LLIF), OLIF does not manipulate the psoas muscle, but orthogonal maneuvers may temporarily stretch the genitofemoral nerve and psoas muscle. However, these complications are very rare compared to LLIF [37,38]. In our study, there were no cases of wound revision or infection in the SP-OLIF group; however, there were also no cases of infection or revision in the C-OLIF group. The reasons for this may be as follows: less damage to the paraspinal muscles and facet joints [39]; little stimulation of the nerve roots due to no laminectomy [37]; and the increased accuracy of pedicle screw insertion using the navigation system.

The SP-OLIF group showed a higher fusion rate at one year after surgery than the C-OLIF group (94.4% vs. 90.0%, *p =* 0.536), which was not significantly different but is consistent compared with previous studies. A meta-analysis by Tai-bang et al. revealed that postoperative fusion rates were similar between the OLIF and TLIF groups, with no statistical difference (mean difference = 1.55, 95% CI: -0.47 to 5.1, *p =* 0.09) [40]. Woods et al. reported that fusion rates of OLIF of L2–5 based on CT at six months was 95.3%, and there was successful fusion at 97.9% of surgical levels [38]. Kotani reported a fusion rate of 96.8% (non-fusion was detected in three patients) in single-position OLIF with PPSF [15]. Several studies report a slightly lower fusion rate of around 90%, and cage sinking and screw loosening often lead to pseudarthrosis [41,42,43]. We believe that correct endplate preparation and proper cage placement can prevent these complications.

This study has some limitations, mainly due to its retrospective design, small sample size, and risk for confounding. A large multicenter randomized controlled trial is needed to obtain higher-level evidence. Our study was performed on L2–5 with single-level OLIF, excluding L5/S1; therefore, there may be differences from previous studies of multi-level OLIF, including L5/S1 or multi-level OLIF. We have considered including these factors in future studies.

## 5. Conclusions

SP-OLIF using the O-arm combined with PPSF serves as an accurate, safe, and effective surgical procedure without the need for a position change compared to C-OLIF using the C-arm. It provides not only a precise view of the anatomy in the unfamiliar position, but also comparable clinical and radiologic outcomes, including fusion rate.

## Figures and Tables

**Figure 1 jcm-12-00312-f001:**
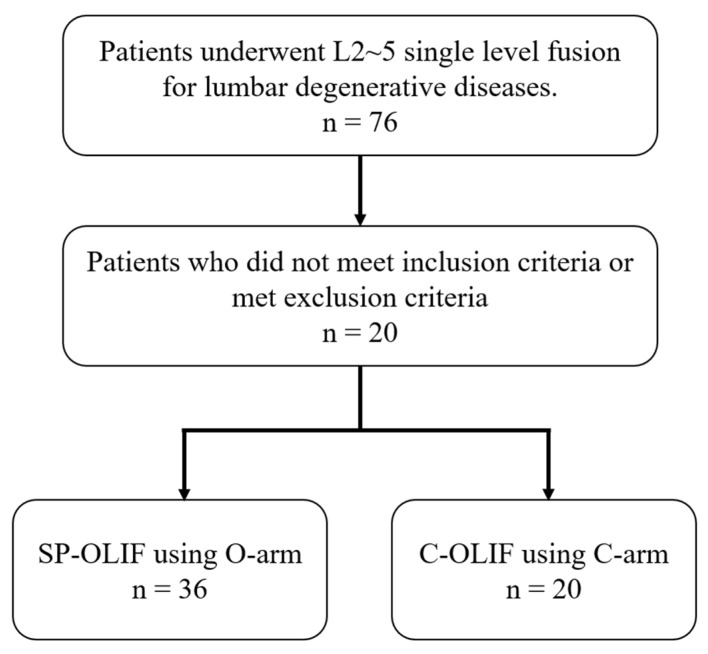
Patient enrollment.

**Figure 2 jcm-12-00312-f002:**
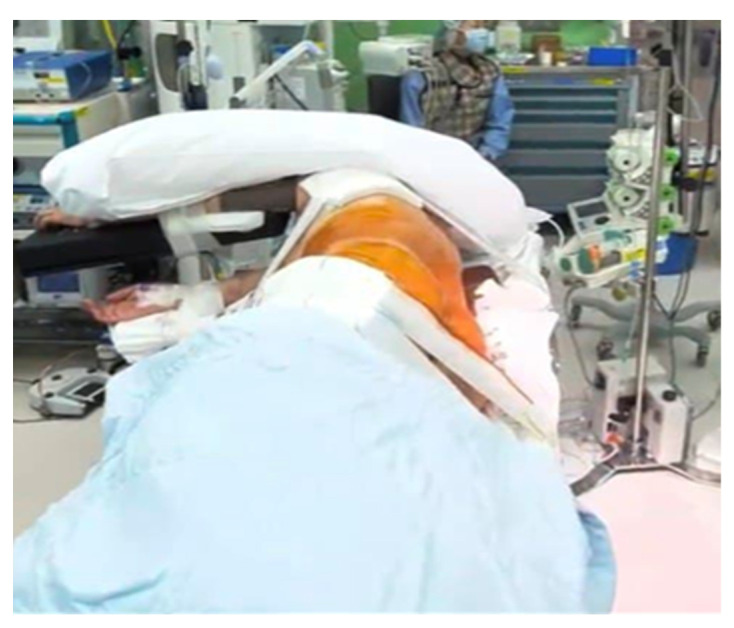
Intraoperative positioning of the patient showing 70-degree right lateral decubitus (left side up).

**Figure 3 jcm-12-00312-f003:**
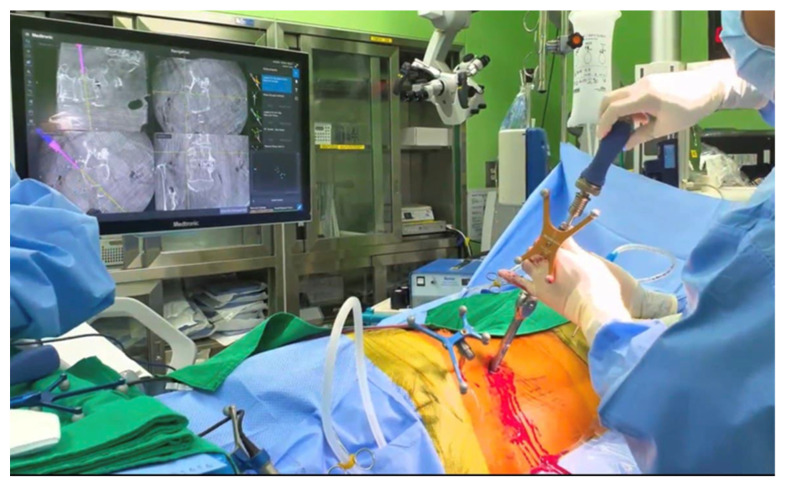
Intraoperative images demonstrating placement of percutaneous pedicle screws under O-arm navigation system.

**Table 1 jcm-12-00312-t001:** Demographic characteristics of the patients.

	SP-OLIF (O-Arm)	C-OLIF (C-Arm)	*p*-Value	95% C.I of the Difference	
				Lower	Upper
Total number	36	20	-		
Sex (male:female)	16:20	6:14	0.255		
Mean age (years)	61.78 ± 8.33	64.30 ± 7.01	0.258	−6.950	1.906
BMI (kg/m^2^)	25.67 ± 5.02	25.00 ± 2.67	0.579	−1.752	3.103
BMD (T-score; mean ± SD)	−0.98 ± 1.03	−1.43 ± 1.38	0.169	−0.199	1.105
ASA class	2.50 ± 0.66	2.20 ± 0.52	0.084	−0.042	0.642
Operative time (mins)	185.00 ± 36.46	198.30 ± 41.75	0.220	−34.773	8.173
Estimated blood loss (mL)	131.94 ± 95.40	270.00 ± 238.64	0.003	−228.103	−48.008
Hospital stays (days)	7.97 ± 2.43	5.40 ± 1.00	0.000	1.429	3.716
Pre-operative diagnosis, n (%)			0.963		
Spinal stenosis	21 (58.3%)	11	-	-	-
Degenerative SPL	12 (33.3%)	7	-	-	-
Spondylolytic SPL	3 (8.3%)	2	-	-	-
Distribution of instrumented levels, n (%)			0.362		
L3/4	4 (11.1%)	4 (20.0%)	-	-	-
L4/5	32 (88.9%)	16 (80.0%)	-	-	-

Values are presented as mean ± standard deviation. SD, standard deviation; BMI, Body mass index; BMD, Bone Mineral Density; ASA, American society of anesthesiologists; SPL, spondylolisthesis.

**Table 2 jcm-12-00312-t002:** Gertzbein-Robbins classification system of pedicle screw accuracy.

Grade	Breach Distance (mm)
A	0
B	<2
C	<4
D	<6
E	>6

**Table 3 jcm-12-00312-t003:** Lumbar pedicle screw placement accuracy grades according to the Gertzbein-Robbins classification system.

GRS Grade	SP-OLIF (O-Arm)	C-OLIF (C-Arm)	*p*-Value
Grade A	139/144 (96.5%)	77/80 (96.3%)	0.915
Grade B	5/144 (3.5%)	3/80 (3.8%)	0.915
Grade C	0/144 (0%)	1/80 (1.3%)	0.179
Grade D, E	0/144 (0%)	0/80 (0%)	1.000

GRS, Gertzbein-Robbins classification system of pedicle screw accuracy.

**Table 4 jcm-12-00312-t004:** Radiologic outcomes.

	SP-OLIF (O-Arm)	C-OLIF (C-Arm)	*p*-Value
Flexion (°)	9.94 ± 5.64	8.05 ± 3.11	0.171
Extension (°)	11.39 ± 5.96	9.98 ± 3.01	0.324
Dynamic (flexion minus extension, °)	1.45 ± 3.98	1.95 ± 1.53	0.598
The number of BTB formation (by CT)	33/36 (91.7%)	17/20 (85.0%)	0.930
Fusion rates	94.4% (34/36)	90.0% (18/20)	0.536

Postop, postoperative; BTB, bridging trabecular bone.

**Table 5 jcm-12-00312-t005:** Clinical outcomes.

	SP-OLIF (O-Arm)	C-OLIF (C-Arm)	*p*-Value
Pre_VAS	7.31 ± 1.31	7.25 ± 1.94	0.899
Pre_ODI	45.69 ± 14.60	46.55 ± 18.77	0.850
Pre_PCS of SF-36	34.53 ± 16.69	41.15 ± 16.61	0.160
Pre_MCS of SF-36	52.50 ± 18.35	51.46 ± 21.20	0.848
Post_VAS	2.56 ± 2.04	3.30 ± 2.27	0.214
Post_ODI	18.81 ± 10.99	28.20 ± 17.06	0.015
Post_PCS of SF-36	62.22 ± 14.09	47.63 ± 15.03	0.000
Post_MCS of SF-36	74.17 ± 13.82	52.89 ± 15.68	0.000

Values are presented as mean ± standard deviation. Preop, preoperative; Postop, postoperative; VAS, Visual analogue scale; ODI, Oswestry disability index; PCS, Physical component summary; SF-36, 36-Item short form survey; MCS, Mental component summary.

**Table 6 jcm-12-00312-t006:** Surgery-related complications.

Type of Complication	SP-OLIF (O-Arm)	C-OLIF (C-Arm)	*p*-Value
Revision	None (0%)	1 (5%)	0.176
Surgical site infection	None (0%)	None (0%)	1.000
Ipsilateral weakness	1 (2.8%)	None (0%)	0.452
Radicular pain or numbness	3 (8.3%)	2 (10%)	0.788
Overall complication rate	4 (11.1%)	3 (15%)	0.673

## Data Availability

Not applicable.

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
