# Peer review of "Single-Position Oblique Lumbar Interbody Fusion and Percutaneous Pedicle Screw Fixation under O-Arm Navigation: A Retrospective Comparative Study"

_jcm, 2022, doi:10.3390/jcm12010312_

Round 1
Reviewer 1 Report (Previous Reviewer 1)
Dear Authors,
Thank you for answering to my observations, I consider you improved the manuscript.
Author Response
Please see the attachment

Reviewer 2 Report (New Reviewer)
1. The authors need to define the study as prospective or retrospective (in the limitations you state that the study had a retrospective design). This should also be added in the title.
2. Was the study registered in a clinical trials registry?
3. Did you perform a sample size calculation?
4. Confidence intervals are missing!
5. Analyzing radiographic techniques demands consideration of the radiation exposure. There are no data about this (potential harmful aspect).
6. How did you correct for confounder?
7. Were the postoperative computed tomography scans additionally performed as part of the study? Were the patients informed about this? Was a written consent necessary?
8. Which ethics committee approved the study? Under which registration number is the ethics vote registered?
9. There is no indication of the high statistical uncertainty associated with the small number of cases.
10. There is no mention of the high risk of confounders.
11. Please check the adherence to the STROBE statemtent (equator-network.org).
12. Please add a flow chart to illustrate the inclusion/exclusion process
13. Please clarify a few things: Which C-arm did you use? Modern C-arms allow digital volume tomography. Has this been used or did you use fluoroscopy? You did not mention that a more accurate visualization of the material was expected due to the more extensive image information (CT vs. 2D fluoroscopy). More information generally leads to more accurate results. This can be explained primarily on technical basis. Unfortunately, I do not see any added value of your study due to the small sample size. You did not correct the results for relevant confounders (due the small sample size this would have been barely possible...). It remains unresolved (and was not expected) whether the use of an O-arm vs. C-arm contributes to differences in terms of relevant endpoints (short/long term complications, fusion rates, short/long term pain levels, adjacent segement degenerations, loosening, etc.).
Round 2
Reviewer 2 Report (New Reviewer)
All comments have been addressed.
Author Response
Thanks to your wonderful comments, our paper has been improved.
As we said in the previous answer, we will submit good research results and papers through a more developed research plan in future research.
Thank you very much for your hard work.
This manuscript is a resubmission of an earlier submission. The following is a list of the peer review reports and author responses from that submission.
Round 1
Reviewer 1 Report
The authors provided a well-documented, almost perfect written and designed study regarding lumbar interbody fusion and percutaneous pedicle screw fixation. The theme and technique used is of interest in European facilities and not only.
I would gladly accept the manuscript if the authors decide on making the minor changes suggested below:
Abstract: when reading the abstract, readers might be interested in knowing what are the follow-up times of the study. Please consider mentioning.
It seems like the images provided have been screenshotted from a video and they lack good quality; is there any way of providing proper images with increased quality?
I think the discussion section should begin with the main finding of your study and afterwards continue with the story flow as you did.
Reviewer 2 Report
These authors present a retrospective review of 36 consecutive patients who underwent single position lateral fusion using an OLIF technique combined with percutaneous posterior pedicle screws and O-arm technology. They conclude based on their findings that this is an effective method for circumferential fusion with high fusion rates, low complication rates, and improved patient outcomes.
First, the concept of the efficacy and safety of single-position lateral fusion is well established, with at least 10 studies in the past year reporting on the efficacy of this technique. Thus this is not a novel concept. The authors do advocate for O-arm assistance, and the benefits of navigated screw placement has also been well published in the literature.
Additionally, without any control arm of this study it is impossible to draw meaningful conclusions regarding their data and how it applies to a more historical method of fluoroscopic-assisted single position lateral. Given the amount of literature in place regarding the accuracy of navigation as well as the efficacy of single-position lateral surgery, I'm not sure this manuscript adds much to the literature already published on the topic.
Round 2
Reviewer 2 Report
I commend the authors on their acknowledgement of the limitations and providing an updated review. They report that their findings are similar to the previous data on this subject, and utilize control group of prior literature. Due to the fact they do not find any differences to prior studies examining the same techniques and the lack of a cohort to directly compare to, I do not believe this paper is unique enough to merit publication. I would suggest they either shift focus to a positioning technique paper (as they mention their technique is different than most) or try to examine variables not previously reported and compare to an internal control cohort.